# Mechanism of hormone and allosteric agonist mediated activation of follicle stimulating hormone receptor

Jia Duan[1,2,13], Peiyu Xu[1,13], Huibing Zhang[3,4,5,6,13], Xiaodong Luan[7,8,9,10,13], Jiaqi Yang[1], Xinheng He[1,2], Chunyou Mao[3,4,5,6], Dan-Dan Shen[3,4,5,6], Yujie Ji[1,2], Xi Cheng[1], Hualiang Jiang[1,2,11,12], Yi Jiang[11,12], Shuyang Zhang[7,8,9,10]✉, Yan Zhang[3,4,5,6]✉ & H. Eric Xu[1,2,12]✉

Follicle stimulating hormone (FSH) is an essential glycoprotein hormone for human reproduction, which functions are mediated by a G protein-coupled receptor, FSHR. Aberrant FSH-FSHR signaling causes infertility and ovarian hyperstimulation syndrome. Here we report cryo-EM structures of FSHR in both inactive and active states, with the active structure bound to FSH and an allosteric agonist compound 21 f. The structures of FSHR are similar to other glycoprotein hormone receptors, highlighting a conserved activation mechanism of hormone-induced receptor activation. Compound 21 f formed extensive interactions with the TMD to directly activate FSHR. Importantly, the unique residue H615[7.42] in FSHR plays an essential role in determining FSHR selectivity for various allosteric agonists. Together, our structures provide a molecular basis of FSH and small allosteric agonist-mediated FSHR activation, which could inspire the design of FSHR-targeted drugs for the treatment of infertility and controlled ovarian stimulation for in vitro fertilization.

Follicle-stimulating hormone (FSH) is secreted by pituitary glands and functions as a stimulator of the estrogen production, ovarian follicles maturation and spermatogenesis[1,2]. FSH acts through FSHR, which is mainly expressed in gonads (ovaries in female and testes in male)[3]. Mutations in human demonstrate that aberrant signaling of FSH-FSHR can lead to serious diseases[4,5], especially infertility and ovarian hyperstimulation syndrome (OHSS). Recently, much more evidence have been found that the function of extra gonadal FSHR is also closely relevant to Alzheimer's disease, osteoporosis, obesity, and cancers[6–9]. In clinical, FSH is a main therapeutic drug used in anovulatory infertility and assisted reproductive technologies for women and hypogonadotropic hypogonadism for men[10,11]. To date, several crystal structures of FSH-bound FSHR fragments, including the leucine-rich repeat hormone-binding domain (FSHR_HB) and the complete ECD fragment (FSHR_ECD), have been reported[12,13], offering an opportunity for us to understand the detailed interactions between FSH and FSHR.

[1]State Key Laboratory of Drug Research, Shanghai Institute of Materia Medica, Chinese Academy of Sciences, 201203 Shanghai, China. [2]University of Chinese Academy of Sciences, 100049 Beijing, China. [3]Department of Biophysics and Department of Pathology of Sir Run Run Shaw Hospital, Zhejiang University School of Medicine, Hangzhou, Zhejiang, China. [4]Liangzhu Laboratory, Zhejiang University Medical Center, Hangzhou, Zhejiang, China. [5]MOE Frontier Science Center for Brain Research and Brain-Machine Integration, Zhejiang University School of Medicine, Hangzhou, Zhejiang, China. [6]Zhejiang Provincial Key Laboratory of Immunity and Inflammatory diseases, Hangzhou, Zhejiang, China. [7]Department of Cardiology, Peking Union Medical College Hospital, Peking Union Medical College and Chinese Academy of Medical Sciences, Beijing, China. [8]Medical Research Center, Peking Union Medical College Hospital, Peking Union Medical College and Chinese Academy of Medical Sciences, Beijing, China. [9]School of medicine, Tsinghua university, Beijing, China. [10]Tsinghua-Peking Center for life science, Tsinghua university, Beijing, China. [11]Lingang Laboratory, 200031 Shanghai, China. [12]School of Life Science and Technology, ShanghaiTech University, 201210 Shanghai, China. [13]These authors contributed equally: Jia Duan, Peiyu Xu, Huibing Zhang, Xiaodong Luan. ✉e-mail: shuyangzhang103@nrdrs.org; zhang_yan@zju.edu.cn; eric.xu@simm.ac.cn

However, the molecular basis of how does FSH activate FSHR is still controversial because of the lack of the full-length FSHR structure.

FSH, along with luteinizing hormone (LH), chorionic gonadotropin (CG) and thyroid-stimulating hormone (TSH), constitute the family of glycoprotein hormones[14], which share similar 3D architectures and function through binding and activation of a subfamily of G protein-coupled receptors, named glycoprotein hormone receptors[15]. There are three members of glycoprotein hormone receptors, FSHR, LHCGR and TSHR, which are activated to mainly couple the Gs protein and lead to up-regulation of the secondary message cAMP in cells[16]. Most recently, the cryo-electron microscopy (cyro-EM) structures of CG-LHCGR-Gs and TSH-TSHR-Gs complexes have also been determined[17–19], further providing insights into hormone specificity and hormone-induced receptor activation. The structures of CG-LHCGR-Gs and TSH-TSHR-Gs complexes suggested that there might be a conserved activation mechanism among glycoprotein hormone receptors.

As a disease-associated drug target, there is an urgent need for highly selective small molecular agonists of FSHR to replace hormone therapy, and the small molecular antagonists of FSHR could also be used for contraception[20,21]. Over the years, there are many small molecular agonists and antagonists towards FSHR that have been developed[22,23]. For example, compound 21f (Cpd-21f)[24] is a small molecule allosteric agonist of FSHR with high affinity and activation efficacy. However, up to now, small molecular agonists and antagonists have not entered clinical research due to their low specificity. The high homology of glycoprotein hormone receptors and the lack of structure information of full-length FSHR continue to hinder the development of small molecule drugs targeting FSHR.

In this study, we report two cryo-EM structures, one is the full-length FSHR bound to FSH, Cpd-21f and Gs protein at a global resolution of 2.82 Å, and the other is the inactive FSHR structure with a global resolution of 6.01 Å. Combined with functional studies, our structures reveal a "push and pull" mechanism of hormone-induced receptor activation, which is highly conserved in glycoprotein hormone receptors. The high-resolution structure of Cpd-21f bound with FSHR also provides the basis for rational design of small molecule drugs with higher specificity. Moreover, the structures of the full-length FSHR also allow us to investigate the mechanisms of activating and inactivating mutations naturally occurred in FSHR-related diseases.

## Results

### Structures of FSH-bound FSHR-Gs complex and inactive FSHR
To obtain a high-resolution structure of FSH-bound FSHR-Gs complex, a constitutively active mutation[25], S273I, was introduced to FSHR to enhance the assembly of the FSH-FSHR-Gs complex, in analogous to the constitutively active mutation S277I in the LHCGR[19]

and S281I in TSHR[17]. Human FSH was incubated with membranes from cells co-expressing FSHR and three subunits of Gs heterotrimer in the presence of Nb35[26], which stabilizes the Gs heterotrimer conformation. Cpd-21f was also added to further stabilize the FSH-FSHR-Gs complex. The final structure was determined at a global resolution of 2.82 Å (Supplementary Fig. 1 and Supplementary Table 1). The EM map was clear to position all components of the complex, including two subunits of FSH, the three Gs subunits, Nb35 and FSHR, which includes the N-terminal leucine-rich repeats 1–11 (LRR1-11), the hinge region, and the C-terminal TMD (Fig. 1a, Supplementary Fig. 2a, b). The local refinement of FSH-FSHR ECD subcomplex yielded a map with a better density at the extended hinge loop from the receptor hinge region (residues 296–328 are not included) (Supplementary Fig. 2c, d). Interestingly, 13 cholesterol and 5 lipid molecules were found surrounding the FSHR TMD, like a belt, which was similar to what has been observed in TSHR structures (PDB: 7XW5)[17] (Fig. 1a, Supplementary Fig. 3).

To obtain stable inactive FSHR proteins for cryo-EM structure determination, a small molecular antagonist, compound 24[27], was added during FSHR expression and purification. Through numerous data collection, we were able to obtain a structure of wild type FSHR in the inactive state at a global resolution of 6.01 Å (Fig. 1b, Supplementary Fig. 4). The EM map was sufficient to place the receptor ECD and TMD, while the density of compound 24 was not observed. Due to the relatively low resolution of the inactive FSHR map, we only built a poly-alanine model for the inactive FSHR structure (Fig. 1b). The data and structural statistics are summarized in Supplementary Table 1.

### Mechanism of FSH-induced FSHR activation
As a canonical member of glycoprotein hormone receptors, FSHR has been chosen as a research model and widely studied in many decades[3,28,29]. The detail interactions between FSH and FSHR were extensively studied based on the two crystal structures of FSH-FSHR_HB and FSH-FSHR_ECD complexes[12,13]. However, two controversial activation models of FSH-induced FSHR activation were also proposed due to the fact that the structures of FSH-FSHR_HB and FSH-FSHR_ECD complexes were determined in dimeric[12] and trimeric states[13,30], respectively. The full-length FSHR structures we solved in the active and inactive states were both in monomeric states. Structural comparison of active FSHR in the FSH-FSHR-Gs complex with FSH-FSHR_HB and FSH-FSHR_ECD complexes, respectively, which reveals that the FSH-FSHR_ECD in our structure is nearly identical to the monomeric FSH-FSHR_ECD structure from the two crystal structures. However, the presence of FSH and Gs protein would prevent receptor dimerization or trimerization in the same convex membrane layer (Supplementary Fig. 5a, b) as observed in the crystal structures. Interestingly, dimerization of the receptor-Gs

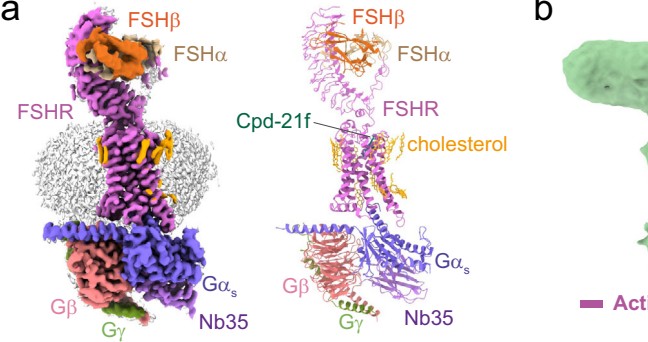

**Fig. 1 | Cryo-EM structures of the FSH-FSHR-Gs complex and the inactive FSHR. a** Cryo-EM density (left panel) and ribbon presentation (right panel) of the FSH-FSHR-Gs complex, the complex density map is shown at level of 0.044. **b** Cryo-EM density (left panel) and ribbon presentation (right panel) of the inactive FSHR, the EM density map is shown at level of 0.017.

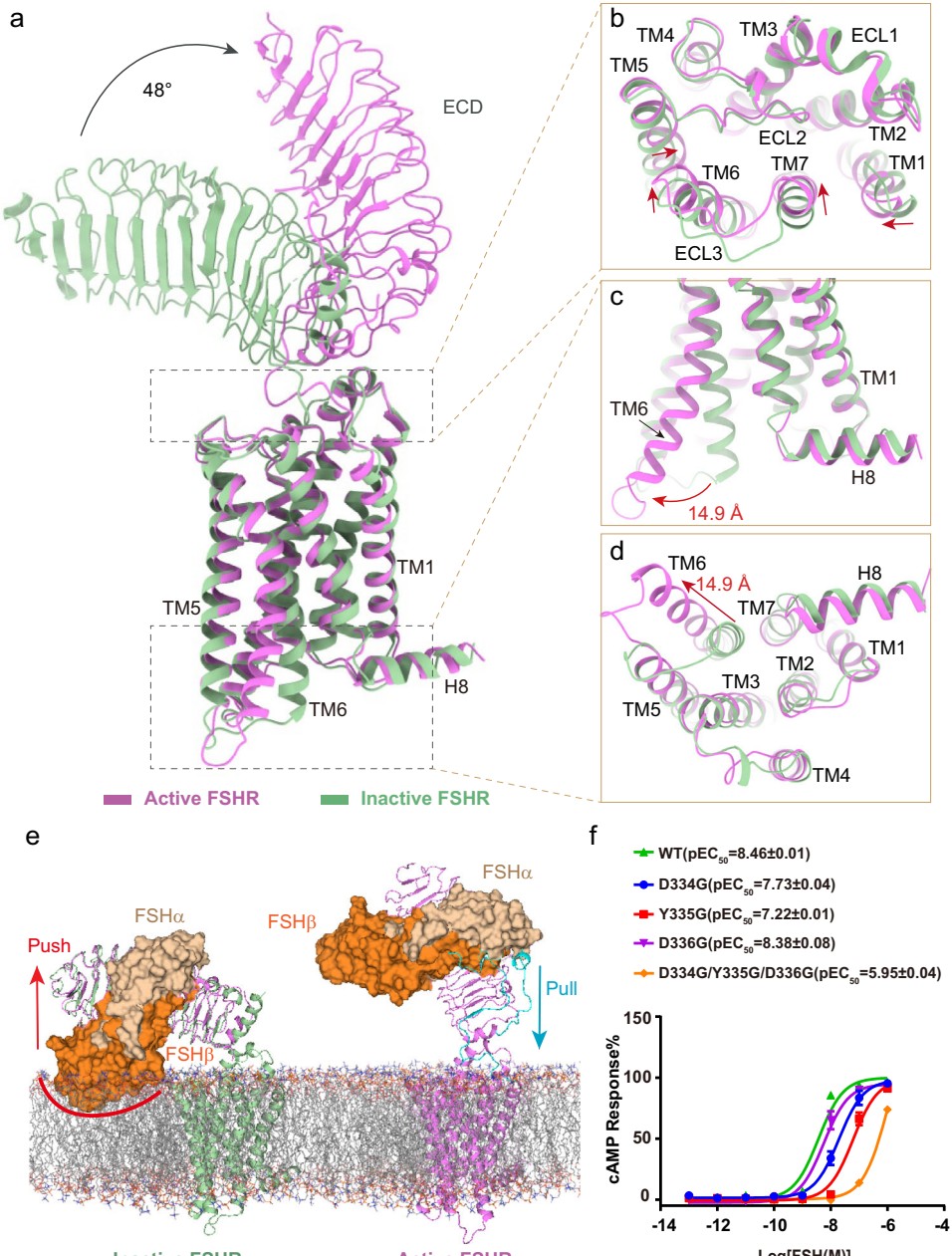

**Fig. 2 | Mechanism of FSH-induced FSHR activation. a–d** Structural comparison of the active and inactive FSHR. The overall structure (**a**), the extracellular side of TMD (**b**), the intracellular side of TMD (**c, d**). **e** The Schematic diagram of FSH-induced FSHR activation, the superposition of ECD structures of the active and inactive FSHR (left panel), and the activation conformation of FSHR (right panel), the clash of FSHβ with the membrane layer is marked with a red line and the extended hinge loop is shown in cyan. **f** Concentration-response curves for FSHR mutants at the extended hinge loop. The representative concentration-response curves from three independent experiments were shown. Data were shown as mean ± S.E.M. from three independent experiments. Source data are provided as a Source Data file.

complex could form from receptors in concave or different membrane layers (Supplementary Fig. 5a).

In the inactive FSHR structure, the ECD conformation is closely tilted toward the membrane layer and the TMD conformation is very similar to the inactive $\beta_{2A}R$[31] (Supplementary Fig. 5c). Conformational changes on FSHR can be seen after FSH binding by structural comparison between active and inactive FSHR. At the extracellular side, FSHR_ECD shows an upward rotation by about 48° after FSH binding (Fig. 2a). Receptor ECD rotation is accompanied by a series of conformational changes in TMD (Fig. 2b–d). At the extracellular ends, TM1, TM6 and TM7 have 2–3 Å inward movement, and TM5 has 1–2 Å inward movement at the middle of the TM helix (Fig. 2b). In contrast, there is a

large movement at the TM6 intracellular end as much as 14.9 Å as measured at the Cα atom of residue S564[6.27] (superscripts refer to Ballesteros-Weinstein numbering[32]) (Fig. 2c, d), which in agreement with the hallmark of class A GPCR activation.

To fully understand how FSH induces FSHR activation, we aligned the ECD structures of the active and inactive FSHR. The superposition of the FSHR ECD structures reveals that the distal region of FSHβ would clash with the membrane layer (Fig. 2e), suggesting a collision push of FSH to rotate the FSHR ECD away from the membrane layer into the upright ECD position in the active FSHR structure, which is similar to CG binding with LHCGR[19], and TSH binding with TSHR[17,18]. On the other side, the extended hinge loop

from FSHR hinge region formed direct interactions with FSH (Supplementary Fig. 2c, d), which is also observed in the crystal structure of FSH·FSHR_ECD complex[13]. Similar hinge loop-hormone interactions were also observed in the structures of both CG-LHCGR-Gs and TSH-TSHR-Gs complexes[17,19]. A conserved sulfated tyrosine at the hinge loop (Y335 from FSHR, Y331 from LHCGR and Y385 from TSHR) was shown to involve in hormone binding and activation of the receptor[17,19,33,34]. Consistently, a single point mutation of Y335G in FSHR resulted in more than a 10-fold reduction of FSH activation potency (Fig. 2f; Supplementary Table 2), while more than a 100-fold reduction of FSH activation potency was caused by three combined point mutations D334G/Y335G/D336G (Fig. 2f; Supplementary Table 2), which further indicated that the extended hinge loop play a role to pull the FSH·FSHR_ECD to rotate away from the membrane layer into the active upright configuration (Fig. 2e), in a similar manner to CG binding with LHCGR and TSH binding with TSHR[17–19].

Structural comparison of the inactive FSHR with the inactive LHCGR and TSHR structures reveals that all three receptors have a similar ECD down conformation (Supplementary Fig. 5d). The ECD of FSHR has a nearly identical position as the LHCGR ECD but a 7°–8° rotation close to the membrane layer relative to the ECD of the inactive TSHR. In addition, the active structure of FSHR also shows high similarities to the active structures of LHCGR and TSHR except subtle differences existing at the receptor ECD configurations (Supplementary Fig. 5e–g). The ECD of active FSHR has an 11.6° rotation away from the membrane layer relative to the ECD of the active LHCGR (Supplementary Fig. 5e). Together, these structural observations support a "push and pull" model for FSH-mediated FSHR activation, which is also conserved in LHCGR and TSHR.

## A conserved structural basis of receptor ECD-TMD signal transmission

Unlike most of class A GPCRs, the glycoprotein hormone-binding sites are located at the distal end of the elongated ECDs of glycoprotein hormone receptors[17–19]. The common "push and pull" model of glycoprotein hormone-induced receptor activation suggests that glycoprotein hormone receptors have evolved a unique paradigm for activation signal transduced from their ECD to TMD, which is distinct from other class A GPCRs. In the active FSHR structure, its ECD contains LRRs 1–11 and a hinge region that comprises a helix termed hinge helix, the extended hinge loop, and LRR12 followed by the conserved P10 region[35], which functions as a tethered intramolecular agonist[36–38] (Supplementary Fig. 2a). The hinge helix is sandwiched between the ECD and TMD, by two conserved disulfide bonds, with one linked to LRR12 and the other linked to the hinge C-terminal P10 region (residues F353-Y362) (Fig. 3a). The links by these two disulfide bonds ensure an integrated and stable conformation of the holo-FSHR structure. Interestingly, the hinge helix in the inactive FSHR structure is elongated and shows different orientation from the hinge helix in the active FSHR structure (Fig. 3b), highlighting that hinge helix plays an important role during FSHR activation. FSHR ECD interacts with the TMD through two major interfaces, which are also observed in LHCGR and TSHR (Fig. 3a, c, e). The first interface is between the hinge helix and the extracellular loop 1 (ECL1) (Fig. 3a, c). The point mutation S273I is located at the N-terminus of hinge helix, where the S273I mutation can enhance the hydrophobic interactions from ECD to TMD, and elevate the activation level of FSHR (Fig. 3c, d). In contrast, the hinge helix in the inactive FSHR structure shows rare interactions with ECL1 by rotating approximately 30° counter-clockwise relative to the hinge helix from the active FSHR structure (Fig. 3b), thus confirming that the interactions between hinge helix and ECL1 is critical for FSHR activation. The second interface is between the P10 region and the top half of TMD (Fig. 3e). As one of the most conserved regions in glycoprotein hormone receptors (Supplementary Fig. 6), the P10 region of FSHR

forms extensive interactions with TM1, TM2, TM7 and the three ECLs (Fig. 3f). The structures of P10 in the three receptors are nearly identical with the RMSD of Cα atoms from FSHR to LHCGR and TSHR are only 0.43 Å and 0.46 Å, respectively (Fig. 3e). Mutational studies of P10 residues in LHCGR and TSHR indicated that most of P10 residues were essential for receptor activation[35,38]. Together, the similar ECD-TMD configuration in all three receptors provided a conserved structural basis of ECD-TMD signal transmission of glycoprotein hormone receptors.

## Basis for FSHR activation by an allosteric agonist

Small molecular FSHR agonists have been proposed to substitute FSH in clinics[21]. Cpd-21f, a 5, 6-dihydroimidazoiso [5, 1-α] quinoline derivative, is an agonist of FSHR[24]. Consistently, Cpd-21f activated FSHR with high affinity and full efficacy (Supplementary Fig. 7a). The EM density map is sufficiently clear to dock Cpd-21f into the FSHR TMD pocket (Fig. 4a), which is created by the top half of TMD, including TM3, TM5, TM6, TM7, ECL2, and ECL3, as well as the N-terminus of P10 region (Fig. 4b, c). Interestingly, the binding site of Cpd-21f was found to be mainly composed of hydrophobic residues (Fig. 4b, d). Our functional data indicated that most of point mutations at pocket residues, except for F353A and M520A, resulted in the decreased activation ability of Cpd-21f to FSHR (Fig. 4e, Supplementary Table 2). Consistently, the binding of Cpd-21f to the FSHR TMD pocket induces inward movements of TM6 and TM7 at the extracellular side (Supplementary Fig. 7b), which directly results in the outward movement of TM6 at the cytoplasmic side of FSHR (Supplementary Fig. 7c), a hallmark of GPCR activation. Compared to LHCGR and TSHR, the location of Cpd-21f is almost completely overlapped with the LHCGR ligand org43553 and the TSHR ligand ML-109 (Fig. 5a), all of which are near the conserved toggle switch residue M585[6.48] (Supplementary Fig. 7c), indicating a conserved activation mechanism of glycoprotein hormone receptors by small molecular allosteric agonists[17,19]. Interestingly, the location of Cpd-21f is also similar to orthosteric agonists seen in other class A GPCRs (Supplementary Fig. 7d), suggesting a similar way for Cpd-21f to directly induce FSHR TMD conformational changes for receptor activation.

FSHR, LHCGR, and TSHR share similar allosteric binding pockets as their high similarities in their TMD sequences and structures, which hinder the development of selective allosteric agonists targeting these important receptors (Supplementary Fig. 6 and Supplementary Fig. 5f, g). Single point and multipoint mutations were performed in order to determine the key residues from the receptors that were responsible for small allosteric agonist selectivity. ML-109 was analyzed first as it was a specific TSHR agonist, which cannot activate either FSHR or LHCGR (Fig. 5c, Supplementary Fig. 7e). Detailed structure analysis indicated that the orientations of the extracellular side of TM6, which forms part of the allosteric agonist binding pocket, were nearly identical between FSHR and LHCGR but different from that of TSHR (Fig. 5b). Compared with TSHR, FSHR and LHCGR have inward shifts at the extracellular side of their TM6, which would clash with the TSHR-selective ligand ML-109 (Fig. 5b). However, swapping of the extracellular portion of FSHR TM6 (residues 591–599) with the corresponding TSHR TM6 region (residues 643–651) did not convert the mutated receptor to respond to ML-109 (Fig. 5c; Supplementary Table 3). In addition, the extracellular portion of TM5 also contributes to the pocket, which has different residues between FSHR and TSHR (Supplementary Fig. 6). However, the replacement of the extracellular portions of both TM5 and TM6 from FSHR (residues 527–536/591–599) with the corresponding TSHR TM5/TM6 regions (residues 579–588 and 643–651) did not increase FSHR response to ML-109 either (Fig. 5c; Supplementary Table 3). Moreover, A352 in FSHR is located at the top of the binding pocket (Supplementary Fig. 7f). The correspondence residue in LHCGR is A349 and in TSHR is E404. However, mutation of A352E plus TM5/TM6 mutations still cannot elevate FSHR response to ML-109 (Fig. 5c; Supplementary Table 3). Surprisingly, mutations

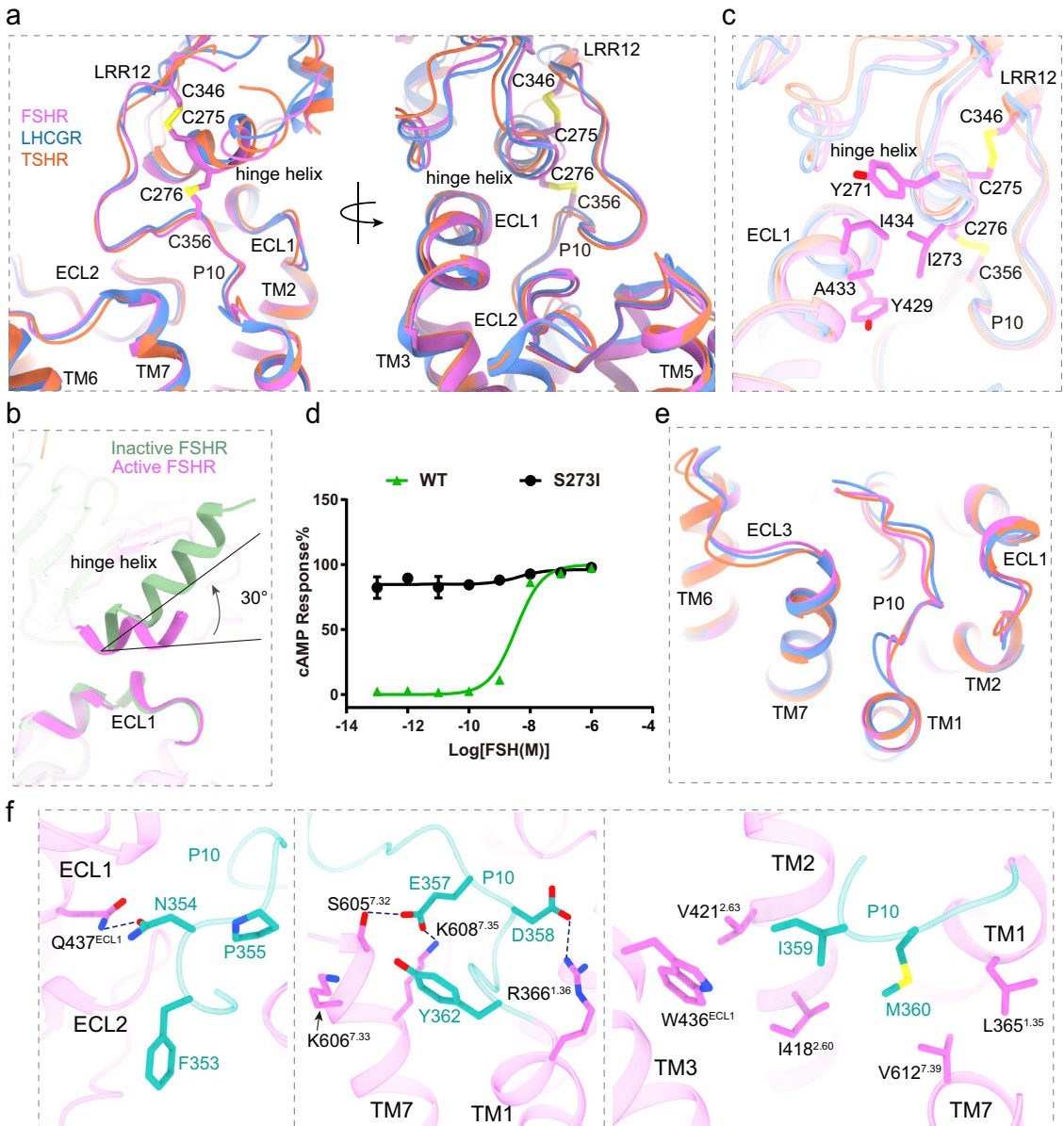

**Fig. 3 | Structural basis for FSHR activation signal transduced from ECD to TMD. a** Structural comparison of FSH-FSHR-Gs complex with CG-LHCGR-Gs and TSH-TSHR- complexes. The similar ECD-TMD configuration in three structures is shown, and the two conserved disulfide bonds in FSHR are shown in sticks. **b** Structural comparison of the hinge helix in the inactive and active FSHR structures. The angle was measured at the Cα atoms of S273 and R283 from the inactive FSHR and C276 from the active FSHR. **c** The first conserved interface between hinge helix and ECL1. **d** Concentration-response curves for WT and S273I mutant receptors. The representative concentration-response curves from three independent experiments were shown. Data were shown as mean ± S.E.M. from three independent experiments. Source data are provided as a Source Data file. **e** The second conserved interface between P10 and TMD. **f** Detailed interactions between P10 and FSHR TMD are shown in sticks.

of H615$^{7.42}$Y or I411$^{2.53}$M in addition to the above TM5/TM6/A352E mutations can both lead to FSHR fully activated by ML-109, and all mutations combined (H615$^{7.42}$Y/I411$^{2.53}$M/ TM5/TM6/A352E) can further increase the activation potency and efficacy of ML-109 to FSHR (Fig. 5d, Supplementary Table 3). On the other hand, single point H615$^{7.42}$Y mutation of FSHR can only be activated by ML-109 weakly (Supplementary Fig. 7g and Supplementary Table 3). Combined mutations of H615$^{7.42}$Y and I411$^{2.53}$M further increased FSHR response to ML-109 (Supplementary Fig. 7g and Supplementary Table 3). H615$^{7.42}$Y and I411$^{2.53}$M are located at the bottom of the pocket and the corresponding residues in TSHR are in direct contacts with ML-109 (Supplementary Fig. 7h). Thus the H615$^{7.42}$Y and I411$^{2.53}$M mutations in FSHR would increase the interactions of the mutated FSHR with ML-109. Consistently, Molecular Dynamic (MD) simulation experiments indicated

that the binding free energy of WT FSHR was higher than FSHR with H615$^{7.42}$Y and I411$^{2.53}$M mutations. Thus, the MD simulations implied that the H615$^{7.42}$Y and I411$^{2.53}$M mutations favor the binding of ML-109 and increase its activating ability for FSHR. ML-109 is able to activate TSHR, the binding energy of ML109 to TSHR is comparable to ML-109 binding to mutant FSHR but not WT FSHR (Supplementary Fig. 8).

Cpd-21f can activate FSHR, LHCGR and TSHR with high potency and efficacy, while the activation potency for FSHR is more than 10-fold higher than LHCGR and TSHR (Supplementary Fig. 7i, Supplementary Table 2 and 4). Detailed structural analysis indicated that the N-ethyl of the Cpd-21f would clash with the Y612$^{7.42}$ in LHCGR and Y667$^{7.42}$ in TSHR, thus weakening the activation potency of Cpd-21f to LHCGR and TSHR (Fig. 5e). Consistently, Mutation H615$^{7.42}$Y in FSHR resulted in nearly 4-fold reduction of activation potency of Cpd-21f to

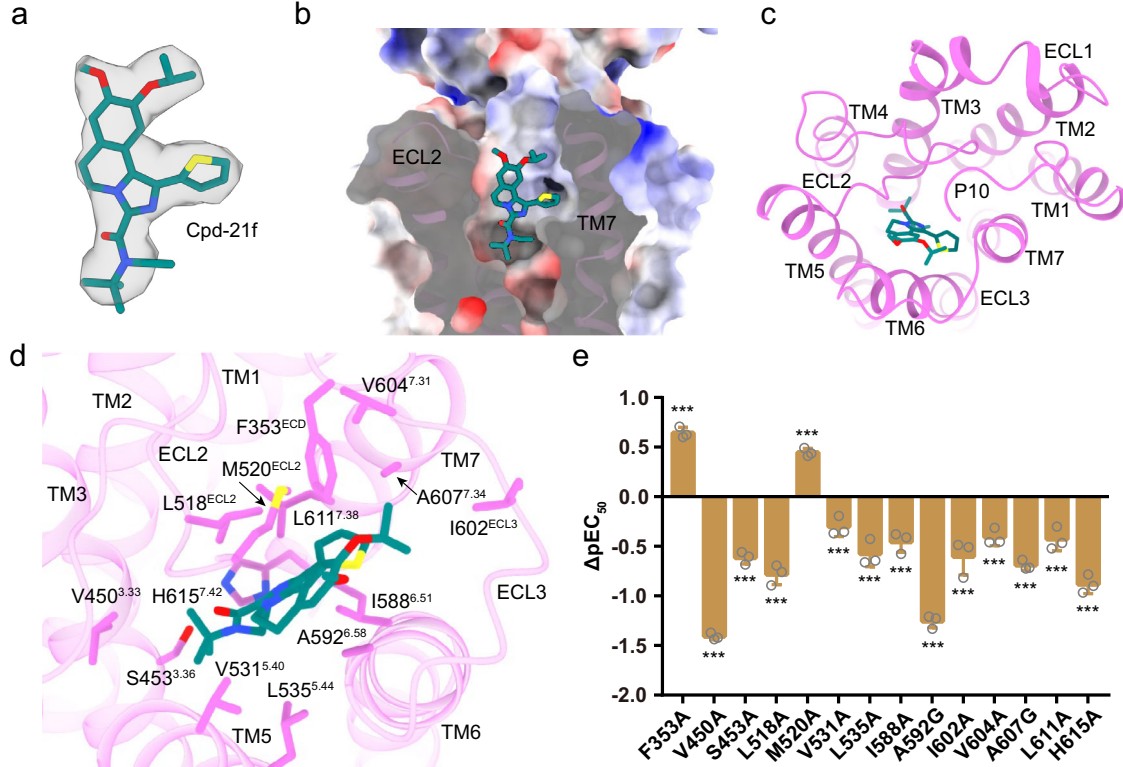

**Fig. 4 | Structural basis for FSHR activation by an allosteric agonist Cpd-21f.**
**a** The structure and EM density of Cpd-21f in the bound structure. The density map is shown at level of 0.17. **b, c** The binding pocket of Cpd-21f in FSHR, from the front view (**b**) and top view (**c**). **d** Detailed interactions between Cpd-21f and FSHR. **e** Effects of different pocket mutations on the potency of Cpd-21f-induced cAMP accumulation. Data were shown as $\Delta$pEC$_{50}$ ± S.E.M. from three independent

experiments, which performed in triplicates, with total repeats of nine for each data point. **$P < 0.01$, ***$P < 0.001$ versus WT. Statistical significance of differences between WT and mutants was determined by two-sided one-way ANOVA with Tukey test. Source data are provided as a Source Data file. All the exact P values are listed as follows: <0.0001, <0.0001, <0.0001, <0.0001, <0.0001, 0.0003, <0.0001, <0.0001, <0.0001, <0.0001, <0.0001, <0.0001, <0.0001, <0.0001.

FSHR (Fig. 5f). Moreover, org 214444-0 was reported as a selective FSHR agonist, which activated FSHR with more than 100-fold higher potency comparing to LHCGR[39]. Org 214444-0 was docked into FSHR TMD binding pocket, which location was also overlapped with Cpd-21f (Supplementary Fig. 7j). The bottom hexahydroquinoline group of Org-214444-0 is larger than the N-tert-Butyl-N-ethyl moiety of Cpd-21f, thus leading to more severe clash of cyclohexanone of Org 214444-0 with Y667[7.42] in TSHR and Y612[7.42] in LHCGR (Supplementary Fig. 7k, l), consistent with the high selectivity of Org 214444-0 for FSHR[39]. Together, our analysis indicated that H615[7.42] in FSHR played essential roles in determining FSHR ligand selectivity and provided new insights for designing more specific small molecular agonists targeting this important receptor for infertility and in vitro fertilization.

### Distribution of diseases associated mutations on the FSHR structure

A distinct feature of glycoprotein hormone receptors is that they have many natural mutations associated with endocrine diseases[40,41]. These mutations are classified into inactivating and activating mutations. In the case of FSHR, the inactivating mutations can cause primary or secondary amenorrhea, premature ovarian failure (POF), and even infertility, while activating mutations can result in OHSS[42]. Comparison with LHCGR and TSHR, FSHR has the fewest natural mutations (Supplementary Fig. 9). There are no more than 20 inactivating mutations and four activating mutations that have been identified in FSHR (Supplementary Fig. 9a)[7]. Interestingly, most of the mutations in FSHR are located at the surface of the receptor rather than the TMD hydrophobic core (Supplementary Fig. 9a), providing a clue that FSHR

TMD is the most constrained structure among glycoprotein hormone receptors.

Based on the active and inactive FSHR structures[43], we modeled the FSHR naturally occurring mutations[7]. Most of inactivating mutations, including P348R, P519I, P587H, are in key structural elements and their mutations could disrupt the stability of FSHR, thus inactivating the receptor. Specifically, P348 is located in the C terminal of LRR12, P587 is located in the receptor TM6 bundle, and P519 is located in the ECL2 (Fig. 6a). On the other hand, the activating mutations, including N431I, T449A/I[3.32], I545T[5.54], and D567G[6.30], are spread in the FSHR TMD (Fig. 6b–e). The N431I mutation is located at the ECL1 near the S273I mutation, which could enhance the hydrophobic interactions between ECL1 and the hinge helix (Fig. 6b), thus elevating the basal activity of FSHR. T449A/I[3.32] mutations are located at the TMD hydrophobic core, which are likely to break the interaction between T449[3.32] and H615[7.42] (Fig. 6c). I545T[5.54] is located near D581[6.44] (Fig. 6d), a PIF motif residue that plays important roles in class A GPCR activation[44]. The I545T[5.54] mutation is likely to reduce the hydrophobic interactions between I545[5.54], I578[6.41] and L460[3.43] to disrupt the packing stability of TM3, TM5 and TM6 (Fig. 6d). Consistently, mutations I542L[5.54] and V597L/F[5.54] at the same locations in LHCGR and TSHR also make the mutated receptor constitutively active (Supplementary Fig. 9b, c). The D567G[6.30] mutation, which is also observed in LHCGR and TSHR, is located at the most conserved position at the N-terminus of TM6 (Fig. 6e, Supplementary Fig. 9b, c), which is likely to disrupt the ionic lock between D567[6.30] and R467[3.50], a highly conserved structure element in the inactive state of class A GPCRs[45].

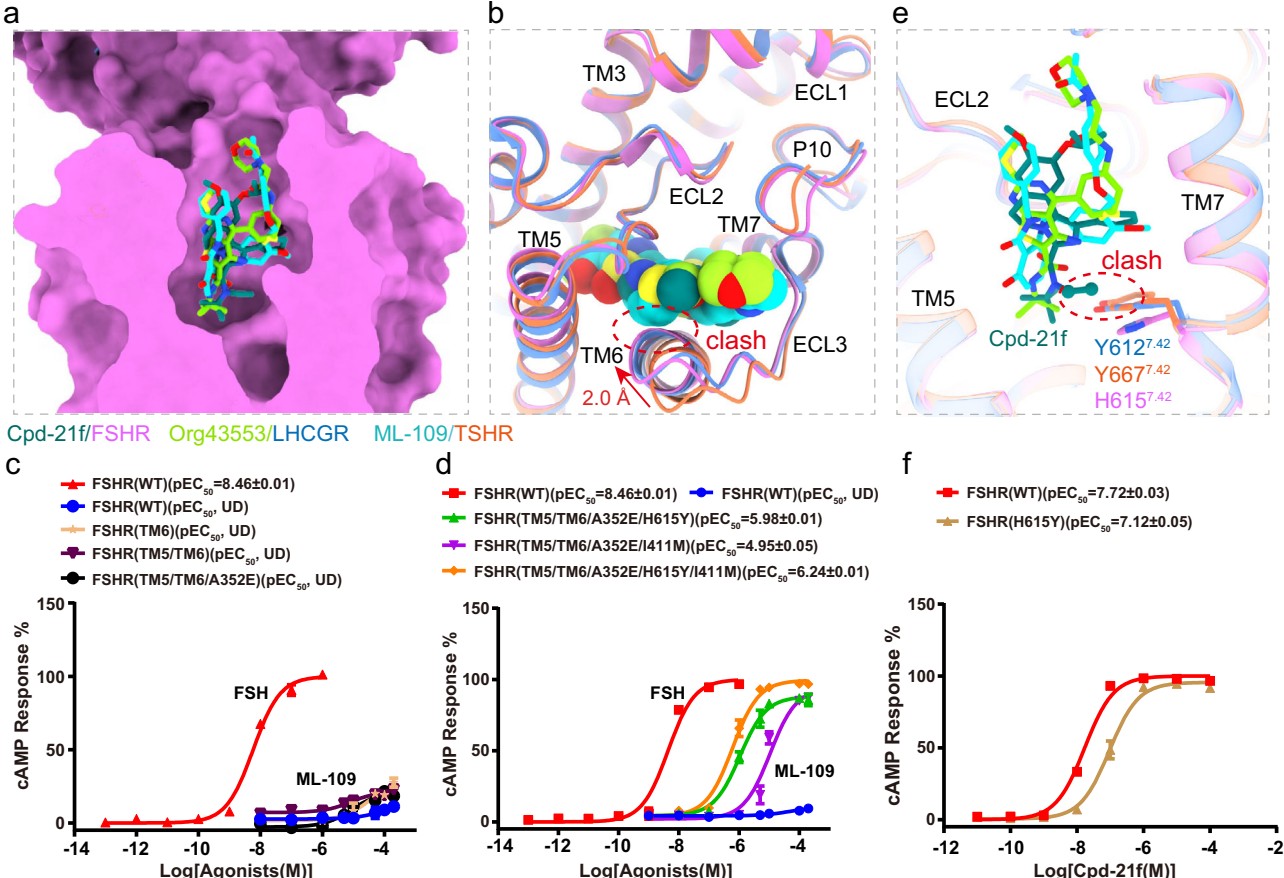

**Fig. 5 | Structural basis of allosteric agonist selectivity for FSHR. a** Structural comparison of Cpd-21f binding pocket in the FSH-FSHR-Gs complex with the Org43553 binding pocket in the CG-LHCGR-Gs complex, and ML-109 binding pocket in the TSH-TSHR-Gs complex. **b** Structural comparison of the binding pockets in FSHR and LHCGR TMDs. The clashes between ML-109 and the extracellular side of TM6 of FSHR and LHCGR are highlighted in red circles. The shift between FSHR TM6 and TSHR TM6 was measured at residues S594 in FSHR and S646 in TSHR. **c, d** Concentration-response curves for WT and mutated FSHR. The representative concentration-response curves from three independent experiments were shown. WT refers to wild type receptor; TM6 means swapping of the extracellular portion of FSHR TM6 (residues 591–599) with the corresponding TSHR TM6 region (residues 643–651); TM5/TM6 means the replacement of the extracellular portions of both TM5 and TM6 from FSHR (residues 527–536/591–599) with the corresponding TSHR TM5/TM6 regions (residues 579–588 and 643–651); TM5/ TM6/A352E means mutation of A352E in addition to the above TM5/TM6 mutations; TM5/TM6/A352E/H615Y means mutation of $H615^{7.42}Y$ in addition to the above TM5/ TM6/A352E mutations; TM5/TM6/A352E/I411M means mutation of $I411^{2.53}M$ in addition to the above TM5/TM6/A352E mutations; TM5/TM6/A352E/H615Y/I411M means mutations of $H615^{7.42}Y$ and $I411^{2.53}M$ in addition to the above TM5/TM6/A352E mutations. **e** Structural comparison of the binding pockets in FSHR, LHCGR and TSHR. The clashes between Cpd-21f and residue $Y612^{7.42}$ of LHCGR and $Y667^{7.42}$ of TSHR are highlighted in red circle. **f** Concentration-response curves for WT and mutated FSHR. The representative concentration-response curves from three independent experiments were shown. UD, undetectable. Data were shown as mean ± S.E.M. from three independent experiments. Source data are provided as a Source Data file.

## Discussion

Glycoprotein hormones and their receptors play central roles in regulating body development, reproduction and metabolism[46,47]. Being an essential part of endocrine system, they are involved in many endocrine diseases such as infertility, OHSS, male-limited precocious puberty[5,48], Graves' and Hashimoto's diseases[49,50]. In this paper, we report the structures of full-length FSHR in both inactive and active states. The active structure of FSHR bound to its natural hormone FSH and a synthetic allosteric agonist Cpd-21f. Detailed analysis of these two structures reveals that FSH binding to FSHR induces a 48° rotation of its ECD toward the upright active conformation, analogous to the 45° and 38° rotations seen in the CG-LHCGR and TSH-TSHR structures, respectively. Consistently, the activation signal transduction from FSHR ECD to TMD mainly depends on the two ECD-TMD interaction interfaces (P10-TMD and hinge helix-ECL1), which are also highly conserved in the LHCGR and TSHR structures. Together, these observations highlight a universal activation mechanism of glycoprotein hormone receptors.

The small molecule agonist of FSHR has been considered as potential oral replacement of protein hormone for infertility treatment. The active FSHR structure with Cpd-21f reveals the binding site of this allosteric agonist in the top half of the FSHR TMD. Structural comparison with LHCGR and TSHR indicates a similar binding pockets for the allosteric agonists, while the unique residue $H615^{7.42}$ in FSHR plays critical roles for FSHR activation by allosteric agonists, providing inspirations for future design of highly selective FSHR ligands. In addition, the FSHR structures also provide molecular basis for the diseases associated mutations.

To date, the structures of FSHR as well as LHCGR and TSHR in both inactive and hormone-bound active states have been solved. In addition, the structures of small molecular allosteric agonists bound with their corresponding receptors also have been solved. Based on our structural observations and functional studies, we proposed a universal model for glycoprotein hormone receptors activation by glycoprotein hormones. In the basal state, the ECD of glycoprotein hormone receptors are tilted toward the membrane layer. Hormone

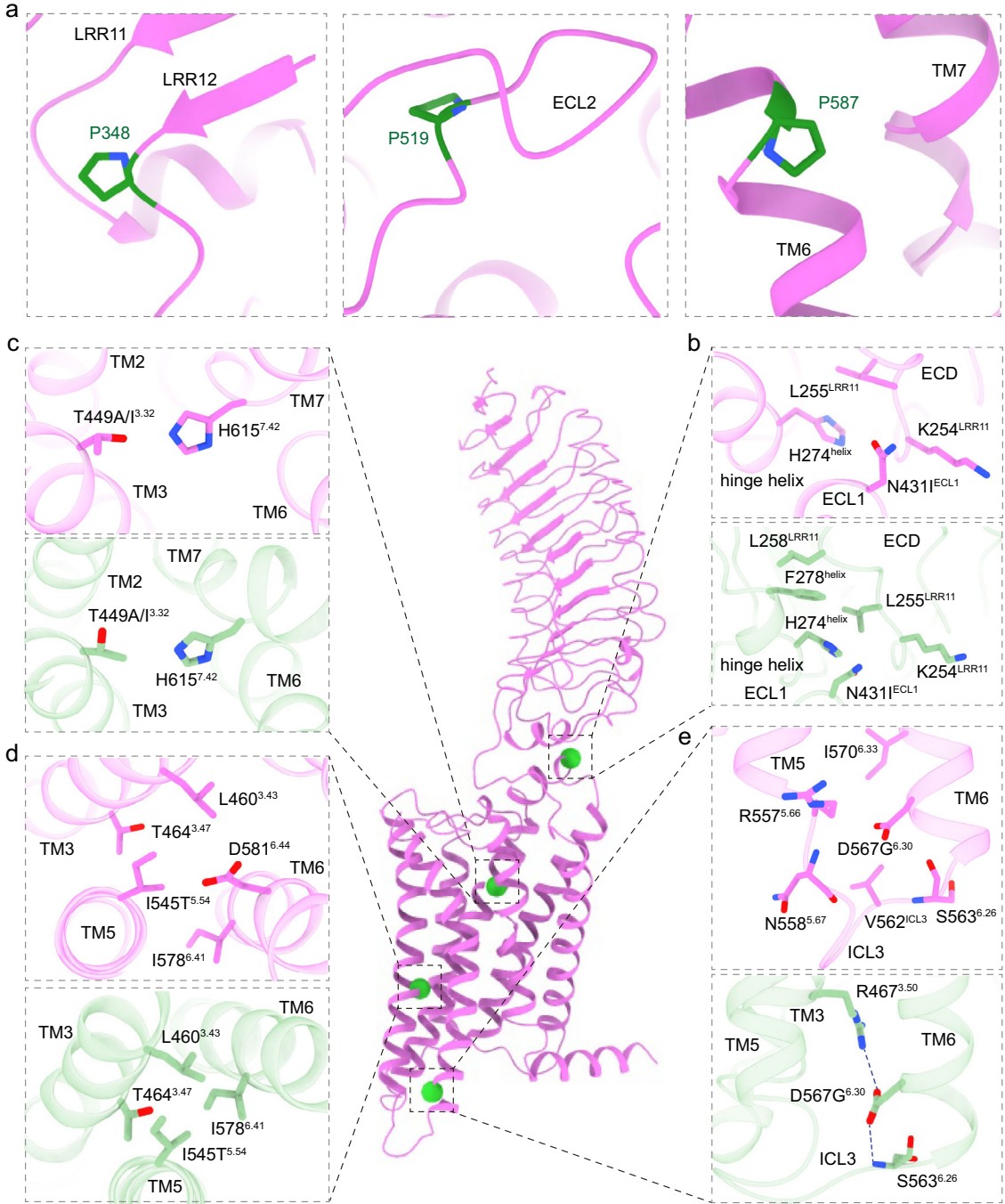

**Fig. 6 | Detailed structural analysis of mutations in FSHR. a** Three inactive mutations in FSHR. **b–e** The detailed interactions surrounded residues N431I[ECL1] (**b**), T449A/I[3.32] (**c**), I545T[5.54] (**d**) and D567G[6.30] (**e**) mutations. The active FSHR structure is shown in violet. The inactive FSHR structure, shown in light green, was modeled by swiss model based on the highest similar template structure. The four active point mutations are highlighted in green spheres.

binding induces the rotation of ECD to the upright position to avoid the clash between hormones and membrane layer. The extended hinge loop from the receptors hinge region interact with hormones to further pull the hormone-ECD complexes in the upright position. On the other hand, the small molecular allosteric agonists can activate glycoprotein hormone receptors alone by direct docking into the receptor TMD pocket, which induces conformational changes in the toggle switch residue and outward movement of TM6 at the cytoplasmic side (Supplementary Fig. 7c). Together, our results not only for FSHR, but also for the whole family of glycoprotein hormone receptors, address the long-standing scientific questions

regarding the mechanisms of hormone-induced receptor activation, disease-associated receptor mutations, and allosteric activation by small molecule agonists.

## Methods
### Constructs
Human FSHR (residues 18–695) was cloned with an N-terminal FLAG and C-terminal His8 tags into a pFastBac vector. The native signal peptide was replaced with the haemagglutinin (HA) to increase protein expression. The constitutively active mutation S273I was introduced to form the FSH·FSHR·Gs complex. A dominant-negative bovine Gαs

construct was generated based on mini-Gs[51]. The three G-protein components, mini-Gαs, rat Gβ1 and bovine Gγ2, were cloned into separately pFastBac vector.

## Expression and purification of Nb35

Nanobody-35 (Nb35) was expressed and purified similarly as previously described[26]. Nb35 was purified by nickel affinity chromatography (Ni Smart Beads 6FF, SMART Lifesciences), followed by size-exclusion chromatography using a HiLoad 16/600 Superdex 75 column and finally spin concentrated to 5 mg/mL.

## Complex expression and purification

FSHR, miniGαs, Gβ1 and Gγ2 were co-expressed in *Sf9* insect cells (Invitrogen) using the Bac-to-Bac baculovirus expression system (ThermoFisher). Cell pellets were thawed and lysed in 20 mM HEPES, pH 7.4, 100 mM NaCl, 10% glycerol, 5 mM MgCl₂ and 5 mM CaCl₂ supplemented with Protease Inhibitor Cocktail, EDTA-Free (Target-Mol). The FSH-FSHR-Gs complex was formed in membranes by the addition of 10 μM Cpd-21f (synthesized in our lab), 1 μM FSH (urinary source, Shanghai Yuan Ye Bio-Technology Co Ltd), 10 μg/mL Nb35 and 25 mU/mL apyrase. The suspension was incubated for 1 h at room temperature. The membrane was then solubilized using 0.5% (w/v) n-Dodecyl β-D-maltoside (DDM, Anatrace), 0.1% (w/v) cholesterol hemisuccinate (CHS, Anatrace) and 0.1%(w/v) sodium cholate for 2 h at 4 °C. The supernatant was collected by centrifugation at 80,000 × *g* for 40 min and then incubated with M1 anti-Flag affinity resin (Genscript) for 2 h at 4 °C. After batch binding, the resin was loaded into a plastic gravity flow column and washed with 10 column volumes of 20 mM HEPES, pH 7.4, 100 mM NaCl, 10% glycerol, 2 mM MgCl₂, 2 mM CaCl₂, 0.01% (w/v) DDM, 0.002%(w/v) CHS, and 0.002%(w/v) sodium cholate, 0.1 μM FSH, 10 μM Cpd-21f and further washed with 10 column volumes of same buffer plus 0.1%(w/v) digitonin, and finally eluted using 0.2 mg/mL Flag peptide. The complex was then concentrated using an Amicon Ultra Centrifugal Filter (MWCO 100 kDa) and injected onto a Superdex200 10/300 GL column (GE Healthcare) equilibrated in the buffer containing 20 mM HEPES, pH 7.4, 100 mM NaCl, 2 mM MgCl₂, 2 mM CaCl₂, 0.05 (w/v) digitonin, 0.0005% (w/v) sodium cholate, 0.1 μM FSH and 10 μM Cpd-21f. The complex fractions were collected and concentrated to 20 mg/mL for electron microscopy experiments.

## Inactive FSHR expression and purification

5 μM compound 24 was added to the culture after *Sf9* insect cells were infected with FSHR baculovirus for 24 h at 27 °C. Cells were then collected after another 24 h later. Cell pellets were thawed and lysed in 20 mM HEPES, pH 7.4, 100 mM NaCl, 10% glycerol, supplemented with 10 μM compound 24 and Protease Inhibitor Cocktail, EDTA-Free (TargetMol). The purification procedures were similar to the FSH-FSHR-Gs complex, except that the procedure of complex formation on membranes was absent.

## cAMP response assay

The full-length FSHR and FSHR mutants were cloned into pcDNA6.0 vector (Invitrogen) with a FLAG tag at its N-terminus. CHO-K1 cells (ATCC, #CCL-61) were cultured in Ham's F-12 Nutrient Mix (Gibco) supplemented with 10% (w/v) fetal bovine serum. Cells were maintained at 37 °C in a 5% CO₂ incubator with 150,000 cells per well in a 12-well plate. Cells were grown overnight and then transfected with 1 μg FSHR constructs by FuGENE® HD transfection reagent in each well for 24 h. cAMP accumulation was measured using the LANCE cAMP kit (PerkinElmer) according to the manufacturer's instructions. The transfected cells were seeded onto 384-well plates with 2500 cells each well, and then incubated with ligands for 30 min at 37 °C, then Eu and Ulight were added separately before cAMP levels were measured. Fluorescence signals were measured at 615 nm

and 665 nm by an Envision multilevel plate reader (PerkinElmer). Data were analyzed using Graphpad Prism8.0, the three-parameter, nonlinear regression equation in Prism suite was used in fitting. The response values were normalized by WT receptors within each individual experiment, with the basal activity for WT as 0, while the fitted E_max of WT as 100. Experiments were performed at least three times, the detail information were attached in the figure legends, each experiment conducted in triplicate. Data were presented as means ± SEM.

## Detection of surface expression of FSHR mutants

The cell seeding and transfection follow the same method as cAMP response assay. After 24 h of transfection, cells were washed once with PBS and then detached with 0.2% (w/v) EDTA in PBS. Cells were blocked with PBS containing 5% (w/v) BSA for 15 min at room temperature (RT) before incubating with primary anti-Flag antibody (diluted with PBS containing 5% BSA at a ratio of 1:150, ABclonal) for 1 h at RT. Cells were then washed three times with PBS containing 1% (w/v) BSA and then incubated with anti-mouse Alexa-488-conjugated secondary antibody (diluted at a ratio of 1:1000, Invitrogen) at 4 °C in the dark for 1 h. After another three times of wash, cells were harvested, and fluorescence intensity was quantified in a BD Accuri C6 software (BD Biosciences) at excitation 488 nm and emission 519 nm. Approximately 10,000 cellular events per sample were collected and data were normalized to WT. Experiments were performed at least three times. The representative flow cytometry data for the detection of receptor surface expression was shown in Supplementary Fig. 10.

## Cryo-EM grid preparation and data collection

For the preparation of cryo-EM grids, 3 μL of the purified protein at 20 mg/mL for the FSH-FSHR-Gs complex, 5 mg/mL for the inactive FSHR, were applied onto a glow-discharged holey carbon grid (Quantifoil R1.2/1.3). Grids were plunge-frozen in liquid ethane using Vitrobot Mark IV (Thermo Fischer Scientific). Frozen grids were transferred to liquid nitrogen and stored for data acquisition.

Cryo-EM imaging of the FSH-FSHR-Gs complex and the inactive FSHR were performed on a Titan Krios at 300 kV in Advanced Center for Electron Microscopy and Cryo-Electron Microscopy Research Center, respectively, Shanghai Institute of Materia Medica, Chinese Academy of Sciences (Shanghai China), A total of 4,025 movies for the FSH-FSH-Gs complex were collected with a Titan Krios equipped with a Falcon4 direct electron detection device at a pixel size of 1.03 Å using the EPU software (FEI Eindhoven, Netherlands). The micrographs were recorded in counting mode at a dose rate of about 15 e/Å²/s with a defocus ranging from −0.8 to −2.0 μm. Each movie was divided into 36 frames during motion correction. For the inactive FSHR protein, a total of 4,495 movies were collected by a Gatan K3 Summit direct electron detector with a Gatan energy filter (operated with a slit width of 20 eV) (GIF) at a pixel size of 1.071 Å using the SerialEM software[52]. The micrographs were recorded in counting mode at a dose rate of about 22 e/Å²/s with a defocus ranging from −1.2 to −2.2 μm. The total exposure time was 3 s and intermediate frames were recorded in 0.083 s intervals, resulting in a total of 36 frames per micrograph.

## Image processing and map construction

For the FSH-FSHR-Gs complex, dose-fractionated image stacks were aligned using MotionCor2.1[53] and CryoSPARC-v3.3[54]. Contrast transfer function (CTF) parameters for each micrograph were estimated by Gctf[55]. Particle selections for 2D and 3D classifications were performed using CryoSPARC-v3.3[54]. Automated particle picking yielded 6,759,790 particles that were subjected to three rounds reference-free 2D classification to discard poorly defined particles, producing 1748,285 particles. The good subsets were further subjected to 6 rounds of heterogeneous refinement, which produced an initial reconstruction for further 3D classification. The selected subsets were subsequently

subjected to non-uniform 3D refinement. The final refinement generated a map with an indicated global resolution of 2.82 Å with 477,950 particles projections at a Fourier shell correlation of 0.143. The particles were further subjected to local-refinement focus on the FSHR-ECD and FSH. The local-refinement generated a map with an indicated global resolution of 3.14 Å at a Fourier shell correlation of 0.143. For the inactive FSHR, image stacks were subjected to beam-induced motion correction using MotionCor2[53]. CTF parameters for non-dose weighted micrographs were determined by Gctf[55]. Automated particle selection yielded 2,781,231 particles using RELION-3.1.2[56]. The particles were imported to CryoSPARC-3.3[54] for 2 rounds of 2D Classification. The well-defined subset accounting for 1,148,175 particles was re-extracted in RELION-3.1.2[56] and was subjected to 2 rounds of 3D classification, producing one good subset accounting for 356,211 particles. The final refinement generated a map with an indicated global resolution of 6.01 Å at a Fourier shell correlation of 0.143.

### Model building and refinement
The crystal structure of human FSH-FSHR_ECD (PDB code: 4AY9), the structure of CG-LHCGR-Gs (PDB code: 7fih) were used as the start for model rebuilding and refinement against the electron microscopy map. The model was docked into the electron microscopy density map using Chimera[57], followed by iterative manual adjustment and rebuilding in COOT[58] and ISOLDE[59]. Real space and reciprocal space refinements were performed using Phenix programs. The model statistics were validated using MolProbity[60]. Structural figures were prepared in ChimeraX[61] and PyMOL (https://pymol.org/2/). For the inactive FSHR, the model of active FSHR from the structure of FSH-FSHR-Gs complex was to generate the initial template. The initial template was docked into the EM density map using Chimera[57]. The model was further rebuilt in coot and real-space refined in COOT[58] and ISOLDE[59]. The final refinement statistics were validated using the module "comprehensive validation (cryo-EM)" in Phenix[62]. The final refinement statistics are provided in Supplementary Table 1.

### Construction of membraned FSHR
FSHR was first oriented by the Orientations of Proteins in Membranes server[63]. Then, the structure was inserted in 100 Å * 100 Å 1-palmitoyl-2-oleoyl-sn-glycero-3-phosphocholine (POPC) membrane according to CHARMM-GUI server[64].

### Molecular docking
The FSH-FSHR-Gs complex was used for docking. The FSHR and small molecules were separated from the complex and prepared in the protein preparation wizard of Schrödinger, Maestro. During the processes, we first assigned bond orders and added hydrogens on the system. Meanwhile, disulfide bonds were created and residue states were generated using Epik at pH = 7.0 ± 2.0. Then PROPKA method was applied to assign protonation state of each residue. At last, the restrained minimization was used with 0.3 Å RMSD constraint on heavy atoms under OPLS4 force field.

### Molecular dynamic simulations
The cryo-EM structure of Cpd-21f-FSH-FSHR and ML-109-TSH-TSHR (PDB ID: 7XW5) complex were used for the construction of the MD simulation systems. Before the simulation, Gs, FSH, TSH, and Cpd-21f were removed from the systems. We first aligned TSHR-ML-109 to FSHR according to the TMD and placed ML-109 in the corresponding FSHR pocket. Then, H615Y and I411M mutations were introduced via PyMOL. In the dual mutation system, we introduced both mutations. We oriented the structures with the Orientations of Proteins in Membranes server and used the CHARMM-GUI server to insert them into POPC (palmitoyl-2-oleoyl-sn-glycero-3-phosphocholine) membrane[63,65]. Then, TIP3P waters were added to the top and bottom of the system. 0.15 Mol/L NaCl

ions and counterions were applied via the distance method[64,65]. FF19SB, LIPID17, and GAFF2 force fields were applied for the parameter of amino acids, lipids, and ML-109, respectively[66,67]. During simulations, the systems were first minimized and equilibrated in the separated 6 processes provided by CHARMM-GUI. Next, 3 parallels of 200 ns production running were applied on the system. The temperature (303.15 K) and pressure (1 atm) were controlled by the Langevin thermostat and isotropic Berendsen barostat, respectively. Long-range electrostatic interactions were treated by the Particle mesh Ewald algorithm and a cutoff of 12 Å was employed for short-range interactions. Between 10 Å and 12 Å, force-based switching was applied for soft changing of interactions. The SHAKE algorithm with hydrogen mass repartitioning was applied to restrain the bond with hydrogens. Hence, the time-step of simulations was 4 fs[68]. The calculation of binding free energy was finished by MMPBSA.py[69]. We used the last 1/4 of trajectories to guarantee that the trajectory had reached equilibrium upon evaluation. The standard deviation was estimated by 3 repeats of simulations.

### Reporting summary
Further information on research design is available in the Nature Portfolio Reporting Summary linked to this article.

## Data availability
The density maps and structure coordinates have been deposited to the Electron Microscopy Database (EMDB) and the Protein Data Bank (PDB) with accession number of EMD-35135, PDB ID 8I2G for the FSH-FSHR-Gs complex; EMD-35136 and 8I2H for the inactive FSHR. The functional data and protein purification data and MD data generated in this study are provided in the Supplementary Information and Source Data files. Source data are provided with this paper.

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

## Acknowledgements

The cryo-EM data were collected at the Cryo-Electron Microscopy Research Center and Advanced Center for Electron Microscopy, Shanghai Institute of Materia Medica (SIMM). The authors thank the staff at the SIMM Cryo-Electron Microscopy Research Center and Advanced Center for Electron Microscopy for their technical support. This work was partially supported by Ministry of Science and Technology (China) grants (2018YFA0507002 to H.E.X.); Shanghai Municipal Science and Technology Major Project (2019SHZDZX02 to H.E.X.); Shanghai Municipal Science and Technology Major Project (H.E.X.); CAS Strategic Priority Research Program (XDB37030103 to H.E.X.); the National Natural Science Foundation of China (32130022 to H.E.X., 32171187 to Y.J., 82121005 to H.E.X. and Y.J., 81922071 to Y.Z.); CAMS Innovation Fund for Medical Sciences (2021-I2M-1-003 to S.Z.); CAMS Innovation Fund for Medical Sciences (2021-CAMS-JZ004 to S.Z.); Tsinghua University-Peking University Center for Life Sciences (045-61020100121 to S. Z.); National Science & Technology Major Project "Key New Drug Creation and Manufacturing Program" of China (2018ZX09711002 to H.J.); Science and Technology Commission of Shanghai Municipal (20431900100 to H.J.); Jack Ma Foundation (2020-CMKYGG-05 to H.J.); Zhejiang Provincial Natural Science Foundation of China (LR19H310001 to Y.Z), the Ministry of Science and Technology (2019YFA050880 to Y.Z.), and the Key R&D Projects of Zhejiang Province (2021C03039 to Y.Z.). Y.Z. is also supported by the Fundamental Research Funds for the Central Universities.

## Author contributions

J.D. designed the expression constructs, purified the FSHR proteins, performed cryo-EM grid preparation and data collection, conducted functional studies, analyzed the structures, prepared the figures, and wrote the manuscript; P.X. and H.Z. performed cryo-EM data calculations, model building, and participated in figure preparation; X.L. supplied the FSH hormone and supervised by S.Z.; Y.-J.J participated in functional studies and protein purification, J.Y. synthesized Cpd-21f and compound 24; C.M. and D.S. participated in cryo-EM data calculations and model building; X.H. performed molecular docking and dynamic stimulations, which were supervised by H.J. and X.C.; Y.J. supervised the studies; H.E.X., Y.Z. and S.Z. conceived and supervised the project and wrote the manuscript with inputs from all authors.

## Competing interests

The authors declare no competing interests.
