## [Peer Review File · Nature Communications]

Mechanism of hormone and allosteric agonist mediated activation of follicle stimulating hormone receptorREVIEWER COMMENTS

Reviewer #1 (Remarks to the Author):

Duan et al. report a cryo-EM structure of the complex of follicle stimulating hormone (FSH)–FSHR–Gs–an allosteric agonist compound 21f. The overall architecture is similar to the previously reported structures of other glycoprotein hormone receptor complexes of CG (chorionic gonadotropin)–LHCGR–Gs and TSH (thyroid-stimulating hormone)–TSHR–Gs. They also reported a low-resolution (~ 6 Å) structure of an antagonist-bound inactive state FSHR. The activation mechanism of FSHR by FSH is similar to the activations of LHCGR by CG and of TSHR by TSH. Furthermore, they performed mutagenesis studies to investigate the binding specificity of the allosteric agonists with the receptors and identified critical residues for allosteric agonist binding. Moreover, they provided structural explanations for the FSHR mutations found in patients.

Overall, the reported full-length FSHR structures provide insights into and advance our understanding of the activation of FSHR, and will be interested to researchers in the field.

Main comments:

1. The authors stated that a universal activation mechanism of glycoprotein hormone receptors mainly depends on two ECD-TMD interaction interfaces (P10-TMD and hinge helix-ECL1). Yet, as outlined in Figure 7, the activation of these receptors by the stimulating antibody and by the allosteric agonist does not involve these two regions. These are only applicable to the activation of these receptors by the glycoprotein hormones.

2. FSH and the small molecule allosteric agonist compound 21f (Cpd-21f) can activate FSHR alone. The authors solved the structure of the complex of FSH–FSHR–Gs–Cpd-21f, but not the structures of FSH–FSHR–Gs and Cpd-21f–FSHR–Gs. Could they use MD simulations to show the activation of FSHR–Gs by FSH or Cpd-21f alone?

Specific comments:

1. Given the low-resolution (~ 6 Å) structure of the inactive state FSHR, they might want to be cautious in describing the minor (such as 1.3 Å) conformational changes during FSHR activation.

2. Could they add MD simulation study to show H615Y and I411M mutations in FSHR indeed increase the activation of FSHR by ML-109?
3. Figure 4 legend: “three independent measurements”: should be more specific: three independent experiments or triplicate?
4. Could they explain the different effects of F353A and M520A from other mutants?
5. Figure 5 c, d and f, as well as Extended Data Figure 7 a, d, f and h: the mean \pm SD of the three independent experiments should be shown.
6. Page 11: “Most of inactivating mutations, including P348R, P519I, P587H, are located in key structural elements and their mutations could disrupt the stability of FSHR, thus inactivating the receptor”. Where is the data to support this statement?
7. Page 13 and Figure 7: “TSHR” should be “glycoprotein hormone receptors”.
8. Table 1: “TSHR” should be “FSHR”.
9. Extended Data Fig. 3: “green” should be “yellow”.
10. Extended Data Fig. 7a: the x-axis label is incorrect.

Reviewer #2 (Remarks to the Author):

The manuscript submitted by Duan et al. reports cryo-EM structures of active (high resolution) and inactive (low resolution) FSHR. Overall, the findings are consistent with recently reported LHCGR and TSHR structures, allowing the authors to propose a general model for the activation of glycoprotein hormone receptors that substantially differs from other known class A GPCRs. The current study also points at some structural peculiarities between the different receptor of the family, in particular in the TMD allosteric pocket that paved the way for the development of more potent and selective low

molecular weight agonists for this family of GPCRs. This work is of great interest for the GPCR community and also opens very important prospects in the field of molecular endocrinology.

Minor remarks:

1) The manuscript is generally clear and well written however it would benefit from an additional round of editing/proof reading for instance:

Line 73: "Over the years, there are many small molecular agonists and antagonists towards FSHR". This sentence does not make sense as it is, something is missing...

Line 256: "...with highly potency and efficacy...". High instead of highly?

2) In the introduction the authors cited potential extragonadal effects of FSH in Alzheimer and cancer. They should also cite potential role in osteoporosis and obesity.

3) A large number of mutations have been engineered in receptors in order to dissect structure-function relationships. In the material and methods, the authors describe a procedure to assess the level of receptor expression at the plasma membrane following transient transfection. This is a very important control as this family of receptor is very susceptible to intracellular trapping upon the introduction of point mutation(s). When reduced potency is observed, is it because of reduced intrinsic efficacy or because less receptor made it to the cell surface? For these reasons, I recommend to show the flow cytometry data with the different mutants vs wild type as supplementary figures.

Reviewer #3 (Remarks to the Author):

This is a beautiful manuscript that not only presents the first active and inactive structures of the full-length FSHR, but also addresses the mechanism of activation and the selectivity of allosteric ligands versus other members of the glycoprotein hormone family.

I would recommend addressing the following issues before publication:

1) The authors report extensive information about the early stages of activation (ECD to TMD or residues around the allosteric ligand). However, I miss a comparison of other structural elements (such as the trigger switch and other activation elements towards the cytoplasmic side) between the inactive and active forms of FSHR here reported but also with other glycoprotein receptors or other class A

receptors to have a complete picture of the activation process. Also, in the discussion, it is written that “The small molecular allosteric agonist can activate glycoprotein hormone receptors alone or in combination with hormones or autoantibody. It is not clear from the current text which activation models the authors propose for the receptor binding FSH alone, the allosteric ligand alone, or both simultaneously.

2) The distances and angles measured to describe changes when comparing structures are illustrative but are ambiguous since it is not stated which atoms or groups of atoms were used, except in one case (page 6 “as much as 14.9 Å as measured at the C α atom of residue S564”).

3) There are two mutants in figure 4c that increase potency. Which is the hypothesis? Both residues seem to be in contact. Is it possible that releasing their interaction facilitates activation with an effect that compensates for the possible deficient binding of the ala mutants?

4) A figure comparing the position of the allosteric ligand with the position of the orthosteric ligands in aminergic or other class A receptors would be valuable.

5) It would be interesting if the authors could comment on why the resolution of the inactive structure is much lower than the active one. Is it simply because of the stabilization provided by the G protein and nanobody? Even though no density for the antagonist was observed, should we expect that the molecule is part of the complex?

Additional points:

- page 18: Are Chimera and UCSC Chimera the same thing?

- page 18: I would guess that “protein preparation wizard” refers to a tool in Maestro. This is not clear.

Manuscript ID: NCOMMS-22-31093-T

Title: Universal mechanism of hormone and allosteric agonist mediated activation of glycoprotein hormone receptors as revealed by structures of follicle stimulating hormone receptor

We thank the reviewers for the positive assessments on the quality and importance of our works. Their constructive suggestions have helped us tremendously in revising our manuscript. In response to their comments, we have performed additional MD simulation experiments to address issues from reviewers. In the following sections, we provide point-by-point responses to the comments by the reviewers of our original paper. The reviewer's comments are in **black** and our responses are in **blue**.

Point-by-point responses to Reviewer #1:

Duan et al. report a cryo-EM structure of the complex of follicle stimulating hormone (FSH)–FSHR–Gs–an allosteric agonist compound 21f. The overall architecture is similar to the previously reported structures of other glycoprotein hormone receptor complexes of CG (chorionic gonadotropin)–LHCGR–Gs and TSH (thyroid-stimulating hormone)–TSHR–Gs. They also reported a low-resolution (~6 Å) structure of an antagonist-bound inactive state FSHR. The activation mechanism of FSHR by FSH is similar to the activations of LHCGR by CG and of TSHR by TSH. Furthermore, they performed mutagenesis studies to investigate the binding specificity of the allosteric agonists with the receptors and identified critical residues for allosteric agonist binding. Moreover, they provided structural explanations for the FSHR mutations found in patients.

Overall, the reported full-length FSHR structures provide insights into and advance our understanding of the activation of FSHR, and will be interested to researchers in the field.

Response: We are very grateful for the positive comments of this work by the reviewer. Each of his/her concerns will be addressed specifically below.

Main comments:

1. The authors stated that a universal activation mechanism of glycoprotein hormone receptors mainly depends on two ECD-TMD interaction interfaces (P10-TMD and hinge helix-ECL1). Yet, as outlined in Figure 7, the activation of these receptors by the stimulating antibody and by the allosteric agonist does not involve these two regions. These are only applicable to the activation of these receptors by the glycoprotein hormones.

Response: We agree with the Reviewer and have removed Figure 7 that describes the stimulating antibody and allosteric agonist.

2. FSH and the small molecule allosteric agonist compound 21f (Cpd-21f) can activate FSHR alone. The authors solved the structure of the complex of FSH–FSHR–Gs–Cpd-

21f, but not the structures of FSH–FSHR–Gs and Cpd-21f–FSHR–Gs. Could they use MD simulations to show the activation of FSHR–Gs by FSH or Cpd-21f alone?

Response: We thank the Reviewer and have conducted four MD simulation systems including FSHR-FSH-Cpd-21f, FSHR-FSH, FSHR-Cpd-21f, and apo FSHR to show the activation of FSHR by FSH or Cpd-21f alone. We run 500 ns × 3 MD simulations to observe the variation of system conformations. Since TM6 is the key helix for activation, we measured the deviation of TM6 (from S564^{6,27} to K598^{6,61}) between snapshots from MD simulations and the cryo-EM complex. The distribution of deviation in each system is shown in below Figure 1.

Response Figure 1. The distribution of TM6 deviation from the cryo-EM structure. The structures were firstly aligned according to the TMD conformation then the deviation of Cas on TM6 residues was calculated.

From Response Figure 1, the FSHR-FSH-Cpd-21f complex maintained a minor difference in TM6 conformation (1.3 ± 0.5 Å) compared with cryo-EM structure, which is in a full G-protein coupling state. The FSHR-Cpd-21f and FSHR-FSH complexes moved the distribution slightly to higher deviation values of 1.4 ± 0.4 Å and 1.7 ± 0.5 Å from the cryo-EM structure. The apo FSHR showed the largest difference of 2.0 ± 0.5 Å from the cryo-EM structure, indicating that the active state of FSHR is not stable in the apo structure as the FSHR-Cpd-21f and FSHR-FSH complexes. Based on the above results, both FSH and Cpd-21f can maintain FSHR in the active state, and the presence of both FSH and Cpd-21f may increase the stability of FSHR in the active state.

Specific comments:

1. Given the low-resolution (~ 6 Å) structure of the inactive state FSHR, they might want to be cautious in describing the minor (such as 1.3 Å) conformational changes during FSHR activation.

Response: We agree with the Reviewer and have removed the descriptions of the minor conformational changes in the figures.

2. Could they add MD simulation study to show H615Y and I411M mutations in FSHR

indeed increase the activation of FSHR by ML-109?

Response: We have conducted five MD simulation systems including FSHR-H615Y-I411M-ML-109 (FSHR Dual), FSHR-H615Y-ML-109, FSHR-I411M-ML-109, WT FSHR-ML-109, and WT TSHR-ML-109 to evaluate the impact of the mutations on ML-109 binding. We run 200 ns \times 3 MD simulations and used molecular mechanics energies combined with the generalized Born and surface area continuum solvation (MMGBSA) algorithm to calculate the binding free energy of ML-109 in each system. We used the last 1/4 of trajectories to guarantee that the trajectory had reached equilibrium upon evaluation. The results were shown in Response Figure 2. The binding free energy of WT FSHR is higher than all mutant systems. Thus, the MD simulations implied that the mutations favor the binding of ML-109 and increase its activating ability for FSHR. Consistently, ML-109 can activate TSHR, the binding energy of ML109 to TSHR is comparable to ML-109 binding to mutant FSHR but not WT FSHR.

Response Figure 2. The binding free energy calculated by MMGBSA in each system.

3. Figure 4 legend: “three independent measurements”: should be more specific: three independent experiments or triplicate?

Response: We agree with the Reviewer and have changed the sentence to “Data were shown as $\Delta pEC_{50} \pm S.E.M.$ from three independent experiments, which performed in triplicates, with total repeats of nine for each data point.”

4. Could they explain the different effects of F353A and M520A from other mutants?

Response: F353A and M520A are the only two mutations that increase instead of decrease of Cpd-21f binding to FSHR. F353 and M520 are located at the entry of the TMD pocket where Cpd-21f binds. Their alanine mutations may increase the open of the pocket thus increasing accessibility of the compound to the pocket.

5. Figure 5 c, d and f, as well as Extended Data Figure 7 a, d, f and h: the mean \pm SD of the three independent experiments should be shown.

Response: We agree with the Reviewer and have added the mean \pm S.E.M in the corresponding figures.

6. Page 11: “Most of inactivating mutations, including P348R, P519I, P587H, are located in key structural elements and their mutations could disrupt the stability of FSHR, thus inactivating the receptor”. Where is the data to support this statement?

Response: From our active FSHR structure (Response Figure 3), we can see P348 is located in the C terminal of LRR12, and P587 is located in the receptor TM6 bundle, which play important roles in stabilizing FSHR conformation, and mutations may disrupt the conformation stability of the receptor. P519 is located in the ECL2, and P519I would change the ECL2 conformation to inactivate FSHR.

Response Figure 3. Locations of P348, P519 and P587 in the active FSHR structure.

7. Page 13 and Figure 7: “TSHR” should be “glycoprotein hormone receptors”.

Response: We thank the Reviewer and have modified the text accordingly.

8. Table 1: “TSHR” should be “FSHR”.

Response: We thank the Reviewer and have modified the text accordingly.

9. Extended Data Fig. 3: “green” should be “yellow”.

Response: We thank the Reviewer and have modified the text accordingly.

10. Extended Data Fig. 7a: the x-axis label is incorrect.

Response: We thank the Reviewer and have modified the text accordingly.

Point-by-point responses to Reviewer #2:

The manuscript submitted by Duan et al. reports cryo-EM structures of active (high resolution) and inactive (low resolution) FSHR. Overall, the findings are consistent with recently reported LHCGR and TSHR structures, allowing the authors to propose a general model for the activation of glycoprotein hormone receptors that substantially differs from other known class A GPCRs. The current study also points at some structural peculiarities between the different receptor of the family, in particular in the

TMD allosteric pocket that paved the way for the development of more potent and selective low molecular weight agonists for this family of GPCRs. This work is of great interest for the GPCR community and also opens very important prospects in the field of molecular endocrinology.

Response: We greatly appreciate the Reviewer for the positive assessment on the importance of this work.

Minor remarks:

1) The manuscript is generally clear and well written however it would benefit from an additional round of editing/proof reading for instance:

Line 73: "Over the years, there are many small molecular agonists and antagonists towards FSHR". This sentence does not make sense as it is, something is missing...

Response: We thank the Reviewer and have modified the sentence to "Over the years, there are many small molecular agonists and antagonists towards FSHR that have been developed."

Line 256: "...with highly potency and efficacy...". High instead of highly?

Response: We thank the Reviewer and have modified the text accordingly.

2) In the introduction the authors cited potential extragonadal effects of FSH in Alzheimer and cancer. They should also cite potential role in osteoporosis and obesity.

Response: We thank the Reviewer and have added the citations accordingly.

3) A large number of mutations have been engineered in receptors in order to dissect structure-function relationships. In the material and methods, the authors describe a procedure to assess the level of receptor expression at the plasma membrane following transient transfection. This is a very important control as this family of receptor is very susceptible to intracellular trapping upon the introduction of point mutation(s). When reduced potency is observed, is it because of reduced intrinsic efficacy or because less receptor made it to the cell surface? For these reasons, I recommend to show the flow cytometry data with the different mutants vs wild type as supplementary figures.

Response: We agree and have added the flow cytometry data with the different mutants vs wild type as supplementary figure.

Point-by-point responses to Reviewer #3:

This is a beautiful manuscript that not only presents the first active and inactive structures of the full-length FSHR, but also addresses the mechanism of activation and the selectivity of allosteric ligands versus other members of the glycoprotein hormone family.

Response: We are very grateful for the positive comments of this work by the reviewer.

I would recommend addressing the following issues before publication:

1) The authors report extensive information about the early stages of activation (ECD to TMD or residues around the allosteric ligand). However, I miss a comparison of other structural elements (such as the trigger switch and other activation elements towards the cytoplasmic side) between the inactive and active forms of FSHR here reported but also with other glycoprotein receptors or other class A receptors to have a complete picture of the activation process. Also, in the discussion, it is written that “The small molecular allosteric agonist can activate glycoprotein hormone receptors alone or in combination with hormones or autoantibody. It is not clear from the current text which activation models the authors propose for the receptor binding FSH alone, the allosteric ligand alone, or both simultaneously.

Response: We agree and have added discussion of conformational changes in the TMD of FSHR between the active and the inactive states. Because the inactive FSHR structure is in low resolution, we used the inactive LHCGR structure to model the inactive FSHR structure. The comparison of the active FSHR structure and the inactive FSHR model reveals similar conformational changes in the toggle switch and other structure elements to LHCGR and TSHR.

Regarding the activation model in the discussion section, we are intended to discuss the activation of FSHR by FSH first, then by allosteric ligand, and then by both simultaneously. We have modified the text to fit the above intention.

2) The distances and angles measured to describe changes when comparing structures are illustrative but are ambiguous since it is not stated which atoms or groups of atoms were used, except in one case (page 6 “as much as 14.9 Å as measured at the C α atom of residue S564”).

Response: We thank the Reviewer and have added the descriptions in the figure legends.

3) There are two mutants in figure 4c that increase potency. Which is the hypothesis? Both residues seem to be in contact. Is it possible that releasing their interaction facilitates activation with an effect that compensates for the possible deficient binding of the ala mutants?

Response: We agree with the Reviewer. F353 and M520 are located at the entry of the TMD pocket where Cpd-21f binds. Their alanine mutations may increase the open of the pocket thus increasing accessibility of the compound to the pocket.

4) A figure comparing the position of the allosteric ligand with the position of the orthosteric ligands in aminergic or other class A receptors would be valuable.

Response: We agree with the Reviewer and have added the comparing figure in Extended Data figure 7. The receptors used include aminergic receptor (5-HT_{1B}), adenosine receptor A_{1A}R, peptide receptors μ OR and AT1R. The comparison reveals that the FSHR allosteric ligand occupies the same space as the orthosteric ligands in other class A receptors.

5) It would be interesting if the authors could comment on why the resolution of the inactive structure is much lower than the active one. Is it simply because of the stabilization provided by the G protein and nanobody? Even though no density for the antagonist was observed, should we expect that the molecule is part of the complex?

Response: There are two reasons below for the lower resolution of the inactive FSHR. First, the active FSHR is bound with FSH and G protein, which is larger in size than the inactive FSHR for single particle cryo-EM structure determination. Second, for the inactive FSHR, antagonist bound to the receptor TMD pocket only to stabilize the inactive conformation of the TMD, while the flexible ECD is not stabilized, which hinders the resolution of the inactive FSHR. In contrast, for the active FSHR, its ECD is stabilized by the FSH, and TMD is stabilized by both Cpd-21f and G protein, which make FSHR in a very stable active conformation for structure determination.

Though no density for the antagonist was observed, we think that the molecule is part of the complex. Because we can't get the apo FSHR protein under our in-vitro purification conditions without the antagonist, we think that antagonist plays an important role in stabilizing the FSHR protein.

Additional points:

- page 18: Are Chimera and UCSC Chimera the same thing?

Response: Chimera and UCSC Chimera are the same. We have changed UCSC Chimera to Chimera.

- page 18: I would guess that "protein preparation wizard" refers to a tool in Maestro. This is not clear.

Response: We have supplied the preparation process in detail as follows: The FSHR and small molecules were separated from the complex and prepared in the protein preparation wizard of Schrödinger, Maestro. During the processes, we firstly assigned bond orders and added hydrogens on the system. Meanwhile, disulfide bonds were created and residue states were generated using Epik at $\text{pH} = 7.0 \pm 2.0$. Then PROPKA method was applied to assign protonation state of each residue. At last, the restrained minimization was used with 0.3 Å RMSD constraint on heavy atoms under OPLS4 force field.

REVIEWERS' COMMENTS

Reviewer #1 (Remarks to the Author):

The authors have satisfactorily addressed most of my concerns.

Reviewer #2 (Remarks to the Author):

The authors addressed my requests and remarks.

Reviewer #3 (Remarks to the Author):

All my points were properly addressed. I recommend publication.

Reviewer #4 (Remarks to the Author):

In this manuscript, Duan and et., obtained cryo-EM structures of FSHR in both active and inactive states, where the active state is bound to FSH and 21f, PAM. They provide a mechanism of activation and PAM selectivity aspects based on their structures and the structures of other members of the glycoprotein hormone receptors. The authors addressed well the previous comments. They have incorporated the results of the MD simulations to show the activation of FSHR-Gs by FSH or Cpd-21f along with an increase of activation in H615Y and I411M mutants. The outcome of the simulations is sound and in line with the experiment. I don't have major comments and consider the manuscript acceptable for publishing.

Irina G. Tikhonova

Manuscript ID: NCOMMS-22-31093A

Title: Mechanism of hormone and allosteric agonist mediated activation of follicle stimulating hormone receptor

We thank the reviewers for the positive assessments on the quality and importance of our works and their recommendation for the publication of our paper.

Reviewer #1:

The authors have satisfactorily addressed most of my concerns.

Reviewer #2:

The authors addressed my requests and remarks.

Reviewer #3:

All my points were properly addressed. I recommend publication.

Reviewer #4:

In this manuscript, Duan and et., obtained cryo-EM structures of FSHR in both active and inactive states, where the active state is bound to FSH and 21f, PAM. They provide a mechanism of activation and PAM selectivity aspects based on their structures and the structures of other members of the glycoprotein hormone receptors. The authors addressed well the previous comments. They have incorporated the results of the MD simulations to show the activation of FSHR-Gs by FSH or Cpd-21f along with an increase of activation in H615Y and I411M mutants. The outcome of the simulations is sound and in line with the experiment. I don't have major comments and consider the manuscript acceptable for publishing.

All four Reviewers have no further comments and have supported for the publication of this paper, for which we are very grateful.